# Comparison between the Effects of Acute Physical and Psychosocial Stress on Feedback-Based Learning

**DOI:** 10.3390/brainsci13081127

**Published:** 2023-07-26

**Authors:** Xiao Yang, Brittany Nackley, Bruce H. Friedman

**Affiliations:** 1Department of Psychology, Old Dominion University, Norfolk, VA 23529, USA; x2yang@odu.edu; 2Department of Psychology, Virginia Polytechnic Institute and State University, Blacksburg, VA 24061, USA; bnackley@vt.edu

**Keywords:** stress, feedback-based learning, cardiovascular reactivity, emotional valence, reward

## Abstract

Stress modulates feedback-based learning, a process that has been implicated in declining mental function in aging and mental disorders. While acute physical and psychosocial stressors have been used interchangeably in studies on feedback-based learning, the two types of stressors involve distinct physiological and psychological processes. Whether the two types of stressors differentially influence feedback processing remains unclear. The present study compared the effects of physical and psychosocial stressors on feedback-based learning. Ninety-six subjects (*M_age_* = 19.11 years; 50 female) completed either a cold pressor task (CPT) or mental arithmetic task (MAT), as the physical or psychosocial stressor, while electrocardiography and blood pressure were measured to assess cardiovascular stress reactivity (CVR). Self-ratings on the emotional valence of the stressors were also obtained. A probabilistic learning task was given prior to and after the stressors. Accuracy in selecting positive (Go accuracy) and avoiding negative stimuli (No-go accuracy) were recorded as learning outcomes. Repeated measures ANOVA and multiple regressions were used to compare the effects of two stressors and examine the effects of CVR and valence on the learning outcomes. The results showed that although the effects of CPT and MAT on feedback processing were not different, CVR and valence influenced Go and No-go accuracy, respectively. The results suggest that stress-modulated feedback-based learning involves multiple pathways and underscore the link between CVR and reward sensitivity. The findings have clinical implications and may contribute to a better understanding of human behavioral systems.

## 1. Introduction

Stress arises in almost every aspect of life and has adverse behavioral and health outcomes, including cardiovascular disease, mental disorders, risk-taking, and suicidal behaviors [1]. Physiological and cognitive mechanisms underlie the outcomes of stress and often interact with each other [2,3,4], which in turn magnifies the negative impacts of stress on behavior and health [5,6]. However, null findings of the effects of stress on cognition have also been reported in recent studies [7,8,9,10]. This inconsistency in the literature suggests that cognitive processes and stressor type may jointly influence observed effects, and different neural pathways may be involved in specific effects of stress on cognition.

Feedback-based learning, a fundamental behavioral process in everyday functioning, may be one mechanism involved in stress outcomes. In feedback-based learning, decisions are made on the basis of the processing of previous positive or negative feedback (i.e., reward or punishment) [11]. Stress has been reported to modulate the processing of feedback [12,13,14,15]. The physiological mechanisms of stress-modulated feedback processing involve activation of the hypothalamic–pituitary–adrenal (HPA) axis, increases in dopaminergic activity, impaired function of the prefrontal cortex (PFC), and enhanced striatal circuits [16,17]. These mechanisms are also thought to contribute to deficient functioning related to aging, along with neurological and mental disorders [11,18,19].

The effects of stress on feedback-based learning have been studied using acute laboratory stressors. For example, exposure to a cold pressor task (CPT) facilitated positive feedback learning [13]. Similarly, Otto et al. reported a selectively enhancing effect of the CPT on positive feedback learning but not on a sophisticated, PFC-related learning process that is independent of feedback processing [14]. Moreover, the Trier Social Stress Test (TSST) [20], a widely used laboratory stressor, has been reported to reduce the use of negative feedback in a learning task [15]. Even the anticipation of a stressor decreased the learning rate of contingencies of choices in a gambling task [21]. Relatedly, stress exposure increases striatal activity, which plays a critical role in the processing of reward [22], and this effect may persist after the stressor is removed [23]. Together, acute stressors tend to have a reciprocal pattern of effects on feedback-based learning: stress improves positive feedback learning but suppresses negative feedback learning.

While various acute laboratory stressors have been used without differentiation in previous studies, concurrent and subsequent physiological and psychological processes may vary across different stressors. There are two major types of stressors: physical and psychosocial stressors, which induce the *defense reaction* (threat to control) and the *defeat reaction* (failure to control), respectively [24]. Physical stressors (e.g., the CPT) initially induce the response of the autonomic nervous system (ANS), and subjective experiences of distress may be considered secondary effects. By contrast, psychosocial stressors (e.g., the TSST) involve top-down processes and influence the body due to their psychological meanings [24]. Therefore, physical and psychological stressors may have type-differentiated effects on the processing of feedback in learning.

To the best of our knowledge, the potential differentiated effects of physical and psychosocial stressors on feedback-based learning have not been examined. This research gap may be due to methodology issues. Given that prior research has focused on the aforementioned physiological mechanisms [16,25], levels of salivary cortisol have often been assessed as a metric of stress responses, and the effects of stress on feedback processing have been related to cortisol responses to stressors [13,14,15]. However, assessments of cortisol levels require a 15 to 30 min time lag between the stressor and the learning task in order to measure the peak cortisol response [26]. This time lag may attenuate the type-differentiated effects of stressors on feedback processing. Moreover, the two types of stressors have different time lengths of stress exposure, which may be a confounding factor. Additionally, while the appetitive–aversive qualities (valence) of stressors may influence stress responses [27,28] and hence feedback processing, these aspects of stressors have received little research attention in the context of feedback-based learning.

### Current Study

The current study was aimed at comparing the effects of physical and psychological stressors on feedback-based learning. A between-group pretest–posttest design was used, in which a modified CPT and a mental arithmetic task (MAT) served as the physical and psychosocial stressors, respectively. The CPT is commonly used in stress reactivity research [29,30]. As a component of the TSST, the MAT induces ANS responses that are related to the activation of the central nervous system [24]. A three-minute MAT is sufficient to induce physiological stress responses [31]. Cardiovascular (CV) indices and self-rated valence of the stressors were used as metrics of stress responses. CV stress responses not only reflect levels of perceived distress [32] but also mirror ANS activity that may differentiate the effects of physical and psychosocial stressors on feedback processing [27,33]. Assessments of CV indices also allow the temporal proximity of the stressors to the learning task. In addition, the use of valence rating can address whether the valence of the stressors influences feedback-based learning.

On the basis of prior reports that acute stressors enhance the processing of reward [13,14,22], we hypothesized that both the CPT and the MA task would enhance positive feedback learning in a subsequent learning task (*Hypothesis 1*). However, we predicted that the physical stressor would suppress negative feedback learning, but the psychosocial stressor would enhance negative feedback learning (*Hypothesis 2*). As noted above, this hypothesis mirrors the possible affective impact of stressors on feedback processing. Moreover, levels of CV stress responses would be positively associated with the enhancing effects of stressors on positive feedback learning (*Hypothesis 3*). In addition, more negative ratings in valence would predict greater facilitating effects of the MA task on negative feedback learning (*Hypothesis 4*). *Hypotheses 3* and *4* reflect the differences in initial reactions and psychological processes between physical and psychosocial stress [24] and the carryover effects of the two types of stressors on feedback learning that are associated with those differences.

## 2. Method

### 2.1. Subjects

One hundred and four subjects were recruited from psychology courses at Virginia Polytechnic Institute and State University. All subjects were non-smokers, and self-reports indicated that they had no history of mental and neurological disorders or cardiovascular disease. They were required to abstain from alcohol for at least 12 h and from caffeine for at least six hours prior to participation [34]. The subjects were randomly assigned to the MAT or CPT group. Two subjects were excluded from analyses because they had prior knowledge about the stimuli used in the present study (see below), which would interfere with task performance. Moreover, six subjects were excluded due to equipment failure. Therefore, the final sample included 96 subjects (*M_age_* = 19.11 years; *SD* = 1.41 years; 50 female). The groups did not differ in age or sex distribution (see Table 1). Subjects received course credits for their participation and were awarded up to USD 20 on the basis of their performance on the learning task. The study protocol was approved by the Institutional Review Board at Virginia Polytechnic Institute and State University.

### 2.2. Feedback-Based Learning Task

A probabilistic selection task was used to examine feedback-based learning. The task was modified from Frank et al.’s probabilistic learning task [11], which assesses the strength of associating a neutral stimulus with positive or negative outcomes, i.e., positive and negative feedback processing, respectively [11,15]. The probabilistic selection task consisted of a learning phase and a testing phase.

#### 2.2.1. Learning Phase

In the learning phase, subjects learned to choose favorable stimuli that were associated with a high probability of reward over less favorable ones. Each trial began with a fixation period with a 1000-ms duration, followed by a stimulus pair. Subjects were asked to press “1” or “3” on a keyboard to choose the stimulus on the left or right side of the screen. The feedback was presented 1500 ms after key pressing, which was either “Correct” with an image of a nickel (i.e., the positive feedback), or “Incorrect, Nothing” (i.e., the negative feedback). If no response was detected within six seconds, the message “No response detected” was displayed on the screen (see Figure 1).

Japanese Hiragana characters served as stimuli in the learning task. Three different stimulus pairs (referred to as *AB*, *CD*, and *EF* hereon) were presented in a random order, and the two stimuli of each pair were also randomized to the left or right side of the screen. The feedback of stimulus selection was probabilistic: choosing *A*, *B*, *C*, *D*, *E*, or *F* led to 80%, 20%, 70%, 30%, 60%, or 40% probability of positive feedback, respectively (see Figure 1). Thus, *A* was the overall best stimulus, while stimulus *B* was the overall worst stimulus. Subjects completed 60 trials (20 trials of each pair) in a block. If the subject selected the stimulus with a. higher probability of positive feedback in 70% or more of *AB* trials, 60% or more of *CD* trials, and 50% or more of *EF* trials in a learning block, then the subject would start the testing phase after the block. If the learning criterion was not reached, another learning block would be given, and so on until the subject reached the criterion. However, the subject would start the testing phase after six learning blocks, no matter whether the learning criterion was reached or not. Two sets of Hiragana characters were randomly assigned as the learning stimuli in the pre- and post-stressor learning task.

#### 2.2.2. Testing Phase

The testing phase was used to determine subjects’ learning outcomes. There was a total of 90 trials in the testing phase, including six trials of each possible combination. The total trials were randomly presented. Learning outcomes were derived from performance in the **Go** trials (*AC*/*AD*/*AE*/*AF*) that included the best stimulus and the **No-go** trials (*BC*/*BD*/*BE*/*BF*) that included the worst stimulus. Subjects were prompted to select the better stimulus in the pair of each trial by “following their gut feeling”. Stimuli were presented for up to six seconds or until key pressing. No feedback was provided in the testing phase.

### 2.3. Cold Pressor Task

A CPT was used as an acute passive coping physical stress. Subjects were asked to immerse their left foot in cold water (~9 °C) for three minutes [35]. This temperature was chosen to avoid sharp pain. The foot CPT would allow subjects to wear the physiological recording device on their upper limbs through the protocol. At the end of the CPT, subjects rated the emotional valence of the task on the Self-Assessment Manikin (SAM) [36], ranging from 1 (*extremely negative*) to 9 (*extremely positive*).

### 2.4. Mental Arithmetic Task

A MAT was used as an acute active-coping psychosocial stressor [24]. The MAT was selected from the TSST and trimmed to three minutes in order to match the time length of the physical stressor [31]. Subjects were asked to count out loud backward from 1000 to 0 in increments of 13 in front of one experimenter. Each time a mistake was made, the experimenter prompted subjects to start from the beginning. Additionally, subjects were told that they were videotaped and that their performance would be used to evaluate their intelligence. The vocal delivery, subjective experience of being evaluated, and cognitive demands in the MAT augment distress and increase cardiovascular responses [37]. At the end of the MAT, subjects rated the valence of the task on the SAM.

### 2.5. Physiological Recording

Electrocardiography (ECG) was measured from CONMED disposable, pre-gelled, stress-testing spot electrodes (ConMed Andover Medical, Haverhill, MA, USA) using a modified lead II configuration. Blood pressure (BP) was monitored continuously from the left arm of the subject by a wrist cuff. Physiological data were collected using a BIOPAC MP150 system (BIOPAC Systems Inc., Goleta, CA, USA). Raw signals recorded from the device were digitized at 1000 Hz (16 bit) and analyzed by BIOPAC AcqKnowledge software 4.4 (BIOPAC Systems Inc., Goleta, CA, USA).

### 2.6. Procedure

After consent was obtained, subjects completed a self-report health history questionnaire to screen for mental and neurological disorders and cardiovascular disease and were then comfortably seated in the laboratory while physiological recording sensors were attached. To measure resting CV indices, subjects were instructed to sit still and watch a 180-s neutral film that depicted aquatic scenes [38], which was followed by a break and the introduction to the learning task. The pre-stress learning task (including the learning and the testing phase) was then delivered. As described in the sections above, subjects learned the probabilities of positive and negative feedback that were associated with three pairs of stimuli in the learning phase. The learning phase consisted of several blocks of learning, and each block included 60 learning trials (see sections above for details). The testing phase started when the subject selected the stimuli with a higher reward probability in each pair for a certain number of learning trials. There was a 20-min break after the pre-stressor learning task in order to wash out the potential effects of the learning task on physiological measures. Afterward, subjects performed the MAT or CPT. Note that subjects were not informed about their group membership until the stressor was delivered. The post-stress learning task was then given, which was followed by a 2-min recovery period. The post-stress learning task had the same format as the pre-stress learning task, while the sets of stimuli in the two learning tasks were different (see sections above). The device was removed from subjects after the recovery period. Subjects were then debriefed and given the monetary rewards that they had earned from their performance on the tasks (see Figure 2).

### 2.7. Data Reduction

Positive and negative feedback learning was evaluated by accuracy in the Go and the No-go trials, respectively. Accuracy was recorded as the rate of selecting the better stimulus. Stress-induced changes in feedback processing were quantified as the post-stress accuracy minus the pre-stress accuracy for the Go and the No-go trials separately.

Heart rate (HR) was derived from ECG signals and defined as the mean beats per minute (bpm) during the resting baseline and stressors. Systolic and diastolic blood pressure (SBP and DBP) were calculated as the mean of the continuous BP measurements during the resting baseline and stressors. HR, SBP, and DBP reactivity were computed as differences in the measurements between the baseline and the stressor, which were then standardized. The cardiovascular reactivity (CVR) score was computed as the sum of the three standardized reactivity scores.

### 2.8. Analytic Approach

One-way analyses of variance (ANOVA) and *χ*^2^ tests were used to compare continuous and categorical variables across groups, respectively. Two separate 2 (condition: baseline vs. stressor) × 2 (type: MAT and CPT) repeated measures ANOVA (rANOVA) tests served to examine the effects of different stressors on positive feedback learning (Go accuracy; *Hypothesis 1*) and negative feedback learning (No-go accuracy; *Hypothesis 2*). The rANOVA was controlled for the order of the pre- and post-stressor learning stimuli. Moreover, we conducted planned comparisons in the form of paired *t*-tests in order to examine the pattern of the influence of stressors on learning outcomes. Effect sizes in the rANOVA and paired *t*-tests were estimated with *η_p_*^2^ and Cohen’s *d*, respectively.

In order to examine the effects of CV stress responses and the valence of the stressors on feedback processing, multiple regressions were conducted in two steps. In the first step, the CVR score and valence score were entered into two separate regression models to predict changes in Go and No-go accuracy (*Hypothesis 3* and *4*). The regression analyses were controlled for stress type and the order of the pre- and post-stressor learning stimuli. In the second step, the interaction terms **CVR × type** and **valence × type** were added to the models to explore the moderation effects of stressor type. Effect sizes of predictors were estimated with partial *R*^2^*_β_*.

## 3. Results

### 3.1. Descriptive Characteristics

As shown in Table 1, the two stressor groups did not differ in BMI, pre- and post-stressor CV measures, or the number of pre- and post-stressor learning blocks. However, the CPT was rated more negatively in emotional valence compared with the MAT.

### 3.2. Effects of Stressors on Learning Outcomes

The rANOVA on Go accuracy indicated a main effect of condition, *F*(1, 93) = 6.58, *p* = 0.012, *η_p_*^2^ = 0.07, but no interaction between task and type, *F*(1, 93) = 0.24, *p* = 0.63, suggesting that stressors increased Go accuracy regardless of type. Planned comparisons revealed that Go accuracy was increased by the MAT, *t*(47) = 0.04, *p* = 0.031, *d* = 0.24, but was not significantly increased by the CPT, *t*(47) = 0.02, *p* = 0.17.

The rANOVA on No-go accuracy indicated no main effect, *F*(1, 93) = 1.88, *p* = 0.17, or interaction between task and type, *F*(1, 93) = 0.44, *p* = 0.511. Moreover, planned comparisons indicated that No-go accuracy was not influenced by the MA task, *t*(47) = −0.02, *p* = 0.25, or the CPT, *t*(47) = 0.03, *p* = 0.49 (see Figure 3).

### 3.3. Effects of Stress Valence and Cardiovascular Reactivity on Learning Outcomes

In the regression model, predictors accounted for a significant proportion of the variance of the Go accuracy change score, *F*(4, 91) = 2.55, *p* = 0.045. Although the Go accuracy change score was not predicted by stress valence, β = 0.01, *SE* = 0.01, *p* = 0.51, higher levels of CVR predicted greater increases in Go accuracy after stressors, β = 0.05, *SE* = 0.02, *p* = 0.036, partial *R*^2^*_β_* = 0.05. However, the model with the interaction terms became nonsignificant, *F*(6, 89) = 1.79, *p* = 0.11, indicating that stress type did not moderate the effect of CVR on the Go accuracy change score.

Similarly, predictors also accounted for a significant proportion of the variance of the No-go accuracy change score, *F*(4, 91) = 2.59, *p* = 0.042, in the model. The No-go accuracy change score was predicted by stress valence, β = 0.03, *SE* = 0.02, *p* = 0.034, partial *R*^2^*_β_* = 0.03, but not by CVR, β = −0.03, *SE* = 0.03, *p* = 0.41. Although the model with the interaction terms was significant, *F*(6, 89) = 2.33, *p* = 0.039, neither CVR × type, β = −0.11, *SE* = 0.07, *p* = 0.11, nor valence × type, β = 0.03, *SE* = 0.02, *p* = 0.19, predicted the No-go accuracy change score.

## 4. Discussion

The primary goal of our study was to compare the effects of physical and psychosocial stressors on feedback-based learning. Moreover, the present study was also aimed at examining the effects of CV stress responses and stress valence on feedback processing. To achieve these goals, a probabilistic selection task was delivered prior to and immediately after a CPT or MAT. The hypotheses were partially supported. Specifically, as predicted, we found that acute laboratory stressors improved positive feedback learning (*Hypothesis 1*). However, *Hypothesis 2* was not supported. Although the effects of the two stressors on feedback processing were not different, CVR and self-rated stress valence were positively related to Go accuracy and No-go accuracy change scores, respectively (*Hypotheses 3* and *4*).

The present results are in line with previous reports that acute stress enhances positive feedback learning [13,14]. The stress-induced enhancement in positive feedback learning has been explained as a result of increased sensitivity to immediate rewards [13,18]. In turn, the increased reward sensitivity reflects a shift from goal-directed to automatic behaviors [14], which may resemble the tendency to be risk-seeking in the loss domain of human decision-making [39]. The increase in reward sensitivity is believed to involve two effects of HPA axis activation on the central nervous system: elevated dopamine activity in corticostriatal circuits and attenuated PFC functioning [16,17]. Our results suggest that the latency of the effects of acute stress on the neural processes is very brief, and the behavioral manifestations of the neural effects occur immediately after stress exposure. However, caution is warranted in generalizing the effects of acute stress on feedback learning to other cognitive processes. For example, episodic memory and crystalized cognition were not influenced by acute stressors [8,10]. The inconsistency between these null findings [8,10] and the present results suggests that acute stressors may differentially impact encoding but not long-term storage or retrieval phases of memory.

The present results failed to differentiate the effects of physical and psychosocial stressors on feedback-based learning, which suggests that acute stress may influence learning through common factors shared by the two types of stressors. As such, perceived distress and HPA axis activation are likely to play central roles in the mechanisms of stress-modulated feedback learning. However, the present results do not rule out the possibility that type-differentiated processes during the two stressors influence subsequent feedback processing. The reciprocal pattern of changes in Go and No-go accuracy (i.e., Go accuracy increases but No-go accuracy decreases after stress exposure) was displayed in the MAT group but not the CPT group [13,15]. Although this pattern did not reach statistical significance (as shown in the results of interaction terms in rANOVA), the results of planned comparisons suggest that there may be indirect effects of mediating factors on learning outcomes [40]. The mediating factors of the relationship between acute stress and feedback processing should be investigated in future studies. In addition, the reciprocal pattern of feedback processing in the MAT group mirrors the overreliance on the reflexive systems of threat processing in acute stress, as indicated by the increased connectivity of the amygdala with the hippocampus, dorsal striatum, and prefrontal areas [41].

Importantly, the present results indicated that greater CVR predicted more enhancement in positive feedback learning. Given that high levels of CV stress responses indicate greater physiological activation during stressors, the relationship between CVR and learning enhancement suggests that the extent of physiological activation during stressors may index increases in dopamine activity in subsequent learning tasks. The present results were in line with prior studies using cortisol as the indicator of stress responses. For example, cortisol responses were negatively correlated with the use of model-based feedback learning that relies on prefrontal functions [14]. The elevation of cortisol level during an acute stressor was positively related to the performance in Go trials in the probability selection task [15]. However, those relationships of cortical responses with changes in learning outcomes were weak, which may be due to the long latency of the peak effect of the stress hormone. Physiological activation has a shorter latency and is easier to monitor than stress hormones. Therefore, ANS indices, such as CV measures, may serve as alternative metrics of stress responses that can be used to study the concurrent effects of stress on feedback-based learning in real life. Moreover, our results also cohere with the putative positive relationship between CV stress reactivity and reward sensitivity [42]. The mental effort of active coping during psychosocial stressors has been conceptualized as the mechanism that links CVR to reward sensitivity [33,42,43,44]. Given that stressor type did not modulate the relationship between CVR and positive feedback processing, our results suggest that the CVR–reward sensitivity relationship may involve mechanisms other than mental effort, such as the perception of stress and the physiological capacity to respond to stressors.

While CVR was related to positive feedback learning, self-rated stress valence influenced negative feedback learning. The present results showed that stressors rated as having a more negative valence attenuated the reduction in negative feedback learning. In other terms, there was a buffer effect of negative valence on decreased use of punishment cues. This finding may be attributed to affective congruence [18]: given that punishment cues in the learning tasks are negative in nature, the negative valence of a stressor matches the punishment cues, which in turn facilitates negative feedback learning. However, this facilitating effect was not exhibited on learning outcomes. This may be because increased reward sensitivity (as noted above) biased attention towards positive feedback and thus reduced the use of negative feedback [15], which offset the facilitating effect of affective congruence on feedback learning. In this regard, the physiological (increased dopamine activity) and psychological aspects (affective congruence) of a stressor interacted with each other.

### 4.1. Implications

Stress-modulated feedback-based learning has been linked to atypical dopamine activity and attenuated PFC function [14,43,44] and may resemble altered valuation systems that underlie aging [13,19] and various mental disorders [45,46,47]. For example, the alteration of valuation systems among older adults may result in social disengagement [48] or a shift from obtaining new resources to preserving existing resources [49]. These effects in turn may lead to behavioral changes that maintain or deteriorate mental and physical health in aging. At a societal level, stressful life events may influence older adults’ financial and healthcare decisions, and those decisions should be considered in improving clinical practice and healthcare systems. Moreover, due to changes in valuation systems, depressed individuals tend to seek sad stimuli to “match” their negative affective state and exacerbate depressive symptoms [50,51]. Those preferences for sad stimuli are associated with current depressive episodes, as opposed to a history of past depression, and are independent of attentional processes [51]. Therefore, temporary functional changes in the dopamine systems occur and affect depressed individuals’ decisions, which may also help explain our present findings. However, given the mixed findings of the effects of stress on cognition in real life [7,9], unlike depressive episodes, the effects of acute stress on feedback processing may not last long enough to directly impact everyday decision-making. Rather, deficits in decision-making indicate altered dopamine activity that results from structural changes in the central nervous system due to prolonged stress [9].

Additionally, our findings of differing effects of CVR and valence on feedback-based learning suggest that stress influences learning through multiple pathways, which contribute to a better understanding of human behavioral systems. Moreover, while heightened CVR has been considered a risk factor for cardiovascular disease [7,52], excessively low levels of CVR may also signal adverse health outcomes [42]. For example, both blunted CVR and thwarted reward sensitivity have been associated with depression [53], substance addiction [54], eating disorders [55], problematic gambling [56], poor self-reported health [57], and impaired cognitive function [58].

### 4.2. Limitations, Future Directions, and Conclusions

The present findings need to be evaluated in light of some design limitations, including potential practice effects on learning and the lack of measures of stress hormones and neural activities. Accordingly, future studies are expected to use different designs and learning tasks to examine the potential type-differentiated effects of stressors on feedback-based learning. Moreover, although the sample size was ample to detect the effect of acute stressors on feedback learning, the current sample might not have been adequate to identify small but meaningful differences in learning outcomes between the two types of stressors. Thus, a larger sample should be included to replicate and confirm the present findings. Another future direction is to study the relationship between CVR and cortisol levels during stressors and their interactive effects on central feedback processing. In addition, the effects of CVR and valence on feedback processing should be further tested with various samples in order to clarify how these aspects of stress responses influence sensitivity to reward and punishment among older adults and people with mental disorders.

In sum, we replicated the previously reported enhancing effects of acute stress on positive feedback learning and found that greater CVR predicted larger increases in positive feedback learning, while more negative stress valence buffered the reduction in negative feedback learning. Although differentiated effects of physical and psychosocial stressors on feedback-based learning were not found, the present findings suggest that the effects of acute stress on learning involve multiple mechanisms. In addition, CVR and self-rated valence provide insight into physiological and psychological processes parallel to HPA axis activation during stressors and have application value in research on stress and feedback-based learning.

## Figures and Tables

**Figure 1 brainsci-13-01127-f001:**
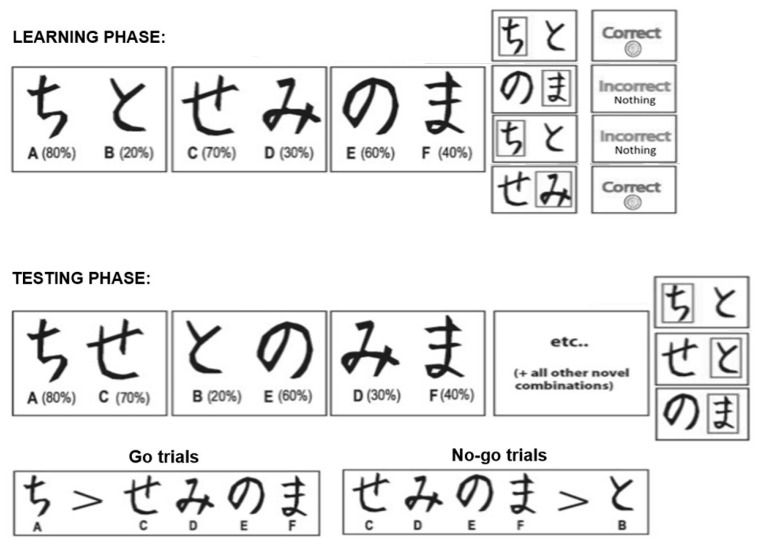
Stimuli in the learning and the testing phase of the feedback-based learning task. Japanese hiragana characters were used in the experiment as the stimuli.

**Figure 2 brainsci-13-01127-f002:**
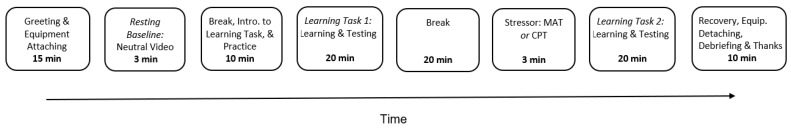
The procedure of the experiment.

**Figure 3 brainsci-13-01127-f003:**
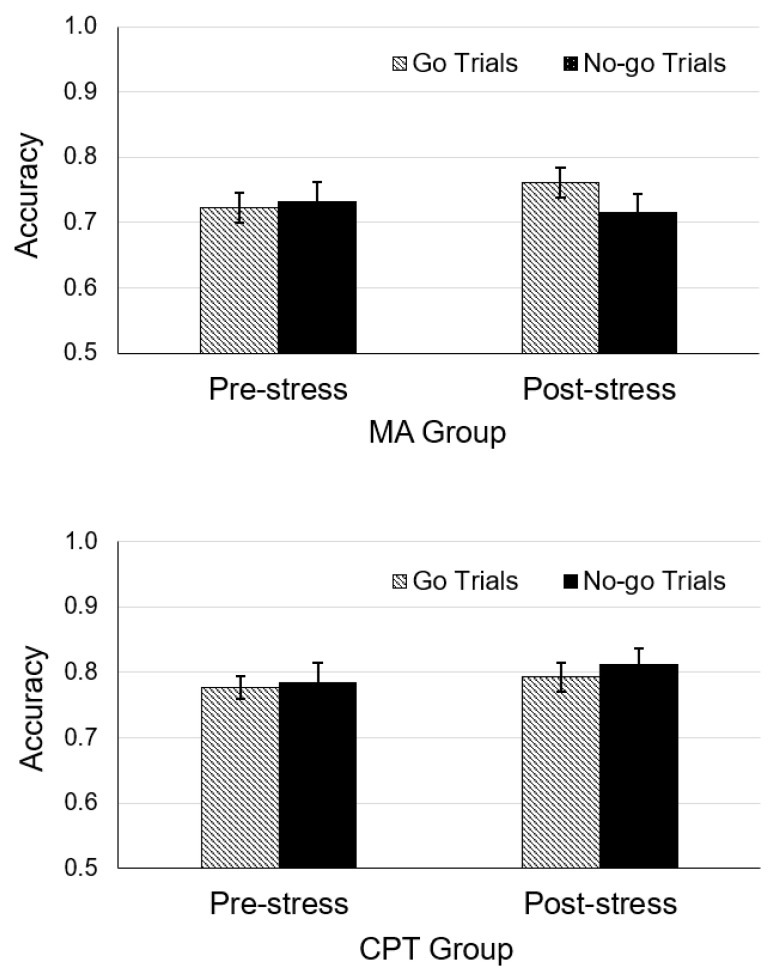
Mean accuracy of Go and No-go testing trials in the mental arithmetic (**MA**) and cold pressor task (**CPT**) groups. Error bars represent ±1 standard error.

**Table 1 brainsci-13-01127-t001:** Descriptive statistics of groups.

Variable	MA Group (*n* = 48)	CPT Group (*n* = 48)	*F* or *χ*^2^ Statistics
Female (*n*)	26 (45.8%)	24 (50.0%)	0.17
Age (years, *SD*)	18.92 (1.18)	19.31 (1.60)	1.90
BMI (kg/m^2^, *SD*)	22.51 (3.25)	23.92 (4.25)	3.32
Stressor Valence	4.05 (1.93)	2.63 (1.86)	13.41 ***
HR (bpm, *SD*)			
Resting	73.55 (10.46)	77.40 (11.09)	3.07
Stressor	79.59 (10.38)	81.25 (10.28)	0.62
SBP (mmHg, *SD*)			
Resting	111.21 (14.33)	111.27 (11.54)	0.01
Stressor	121.91 (18.04)	122.67 (16.41)	0.05
DBP (mmHg, *SD*)			
Resting	67.13 (11.11)	66.96 (9.64)	0.01
Stressor	75.53 (14.74)	75.61 (12.25)	0.01
Number of Learning Blocks Needed (*SD*)			
Pre-stressor	1.98 (0.95)	2.21 (1.01)	1.32
Post-stressor	1.90 (0.87)	2.06 (1.00)	0.75

*Note*: MA = Mental arithmetic; CPT = Cold pressor task; BMI = Body mass index; HR = Heart rate; SBP = Systolic blood pressure; DBP = Diastolic blood pressure. Cardiovascular indices were measured at a resting state and during the stressors. Number of learning blocks was counted in the learning phase of the pre- and post-stressor learning task. All statistics are unadjusted. *** *p* < 0.001.

## Data Availability

The data presented in this study are available on request from the corresponding author. The data are not publicly available due to privacy concerns.

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
