# Peer review of "Comparison between the Effects of Acute Physical and Psychosocial Stress on Feedback-Based Learning"

_brainsci, 2023, doi:10.3390/brainsci13081127_

Round 1

Reviewer 1 Report

This paper reports the results of a study examining the effects of distinct types of acute, time-limited stressors (physical and psychosocial) on performance on a feedback-based learning task.

The paper is well-written and organized in a logical manner. There are no significant flaws in the methodology or the interpretation of the study results, and the authors have provided a clear rationale for their research.

There are certain minor aspects of the paper which would benefit from correction or clarification:

1. In the Introduction, the authors have reported the effects of a specific psychosocial stressor, the Trier Social Stress Test (TSST), on feedback-based learning; in the current study, a mental arithmetic task (MAT) was used. Though these tasks can be broadly "lumped" together as psychosocial stressors, they differ in certain important aspects. Why was the MAT preferred over the TSST in this study? How might the results have differed if the TSST had been used?

2. The authors have mentioned that prior studies have examined salivary cortisol levels when examining the effect of stress on learning. What were the results of these tests? Have any other biological markers been studied in this context? This could be covered briefly in the Introduction, and at more length in the Discussion when the authors present their findings related to  the cardiovascular response to stress.

3. In the description of the study's methods, it is stated that all subjects were "free of histories of mental disorders". How was this ascertained? Was any general screening instrument used to look for current symptoms of anxiety or depression in participants prior to inclusion in the study? Were subjects with medical conditions that could affect cardiovascular responses excluded (or were there no such cases in the subjects recruited)?

4. Did factors such as age, gender and BMI (which were comparable across groups, but which would have differed between subjects) significantly influence the study findings (i.e., in terms of between-subject variability)? Was this specifically analyzed at any stage of this study?

5. Could any other physiological parameters (e.g., skin conductance, heart rate variability, respiratory parameters) have been measured alongside heart rate and blood pressure? Was there any specific reason for focusing only on the latter two measures?

6. At the beginning of the Discussion, it would be useful to state which of the study hypotheses (1-4) were confirmed, and which could not be confirmed / were rejected based on the study results.

7. The Discussion would benefit from comparisons with similar studies from other researchers / groups, even if they have used distinct measures of the physiological stress response or different tasks to induce physical / psychosocial stress.

7. The section entitled "Implications" loses some focus in its last few sentences. As the current study is not specifically designed to examine the relationship between CVR and adverse health outcomes, lines 376-381 seem out of place. In their place, the authors could discuss the implications of their findings for learning in real-life situations, and in relation to specific kinds of stressors.

8. Besides the limitations listed in lines 383ff., the authors should also discuss whether their sample size was adequate to identify small (but meaningful) differences in the effects of each task on feedback-based learning. (In other words, was the failure to confirm any of the hypotheses due to a lack of statistical power?)

Author Response

Reviewer 1’s original comments:

Comment 1: In the Introduction, the authors have reported the effects of a specific psychosocial stressor, the Trier Social Stress Test (TSST), on feedback-based learning; in the current study, a mental arithmetic task (MAT) was used. Though these tasks can be broadly "lumped" together as psychosocial stressors, they differ in certain important aspects. Why was the MAT preferred over the TSST in this study? How might the results have differed if the TSST had been used?

Response: We thank the Reviewer for the comment. The MAT was a component of the TSST. Although the TSST might induce the stronger stress responses, studies have indicated that the MAT alone was adequate to elicit physiological stress responses and subjective experience of distress (Linden, 1991; Spangler & Friedman, 2015). Moreover, we chose the MAT to match the time length of the physical stressor. We added a few sentences in the second paragraph on p.3 and the first paragraph on p.7 to justify the selection of the MAT.

References:

Linden, W. (1991). What do arithmetic stress tests measure? Protocol variations and cardiovascular responses. Psychophysiology28(1), 91-102.

Spangler, D. P., & Friedman, B. H. (2015). Effortful control and resiliency exhibit different patterns of cardiac autonomic control. International Journal of Psychophysiology96(2), 95-103.

Comment 2: The authors have mentioned that prior studies have examined salivary cortisol levels when examining the effect of stress on learning. What were the results of these tests? Have any other biological markers been studied in this context? This could be covered briefly in the Introduction, and at more length in the Discussion when the authors present their findings related to the cardiovascular response to stress.

Response: In the revised manuscript, we described the findings of studies using cortisol as the indicator of stress responses on pp.2-3, and discussed those studies in the first paragraph on p.13.

Comment 3: In the description of the study's methods, it is stated that all subjects were "free of histories of mental disorders". How was this ascertained? Was any general screening instrument used to look for current symptoms of anxiety or depression in participants prior to inclusion in the study? Were subjects with medical conditions that could affect cardiovascular responses excluded (or were there no such cases in the subjects recruited)?

Response: Again, we thank the Reviewers for identifying the issue. In participant recruitment, the exclusion criteria in the present study were described in recruitment materials for potential participants. During the study, we used a self-report health history questionnaire to screen for mental disorders. Moreover, in the same questionnaire, participants were also asked to report their histories of neurological disorders and cardiovascular disease. In the revised manuscript, we added those details in the description of the experiment on p.4 and p.7.

Comment 4: Did factors such as age, gender and BMI (which were comparable across groups, but which would have differed between subjects) significantly influence the study findings (i.e., in terms of between-subject variability)? Was this specifically analyzed at any stage of this study?

Response: The age range (18-21 years) of the sample was small, and age was not correlated with any outcome measures. There were gender effects on heart rate and blood pressures: female participants had higher heart rate but lower (?) SBP and DBP, as compared to male participants, but there was no gender effect on learning performance. In the recruitment, we monitored the gender ratio in the two groups, and the gender ratio was similar in the final samples. While BMI was correlated with heart rate and BP, it did not relate to cardiovascular reactivity or learning outcomes.

Comment 5: Could any other physiological parameters (e.g., skin conductance, heart rate variability, respiratory parameters) have been measured alongside heart rate and blood pressure? Was there any specific reason for focusing only on the latter two measures?

Response: We have calculated heart rate variability (HRV) measures and monitored respiratory activity but did not measure Galvanic skin responses. The selection of heart rate and blood pressure as the indicators as cardiovascular responses to laboratory stressors was based on existing literature (e.g., Kamarck & Lavallo, 2003; Matthews et al., 1993, 2004). As for HRV measures, there is a stationarity issue in assessments of phasic HRV (Berntson et al., 1997). HRV analyses are time series analyses, which assume the data have a stable mean and variance over time (i.e., stationarity). During a task, stimuli may alter subjects’ state and change the mean and variance of heart period data, which violates the assumption of stationarity. We monitored respiratory activity to identify the artifacts in ECG inter-bear interval series, which were not primary dependent variables in the current study. Moreover, the vocal delivery of responses in the MAT introduced the artifacts that impacted a considerable amount of ECG and respiratory signals.

References:

Berntson, G. G., Thomas Bigger Jr, J., Eckberg, D. L., Grossman, P., Kaufmann, P. G., Malik, M., ... & Van Der Molen, M. W. (1997). Heart rate variability: origins, methods, and interpretive caveats. Psychophysiology34(6), 623-648.

Kamarck, T. W., & Lovallo, W. R. (2003). Cardiovascular reactivity to psychological challenge: conceptual and measurement considerations. Psychosomatic Medicine65(1), 9-21.

Matthews, K. A., Woodall, K. L., & Allen, M. T. (1993). Cardiovascular reactivity to stress predicts future blood pressure status. Hypertension22(4), 479-485.

Matthews, K. A., Katholi, C. R., McCreath, H., Whooley, M. A., Williams, D. R., Zhu, S., & Markovitz, J. H. (2004). Blood pressure reactivity to psychological stress predicts hypertension in the CARDIA study. Circulation110(1), 74-78.

Comment 6: At the beginning of the Discussion, it would be useful to state which of the study hypotheses (1-4) were confirmed, and which could not be confirmed / were rejected based on the study results.

Response: We now explicitly stated which of the hypotheses were supported and rejected in the first paragraph in the Discussion section (p.11).

Comment 7: The Discussion would benefit from comparisons with similar studies from other researchers / groups, even if they have used distinct measures of the physiological stress response or different tasks to induce physical / psychosocial stress.

Response: We included additional studies and the discussion of the similarity and discrepancy between the present results and prior findings in the Discussion section (pp.11-13).

Comment 8: The section entitled "Implications" loses some focus in its last few sentences. As the current study is not specifically designed to examine the relationship between CVR and adverse health outcomes, lines 376-381 seem out of place. In their place, the authors could discuss the implications of their findings for learning in real-life situations, and in relation to specific kinds of stressors.

Response: In the revised manuscript, we emphasized on the implications of our findings in aging and depression in this section (pp.14-15).

Comment 9: Besides the limitations listed in lines 383ff., the authors should also discuss whether their sample size was adequate to identify small (but meaningful) differences in the effects of each task on feedback-based learning.

Response: The limitation of the sample size to detect the small effect in differences in learning outcomes between the two types of stressors is acknowledged in the second paragraph on p.15.

Reviewer 2 Report

This is an interesting article.

Some points that need clarification is the recruitment process of the sample.

The procedure is not clear to the reader as well as the included material, so please describe in more detail.

In addition to that, the introduction and discussion sections need more references to support their points. Although there is a plethora of research supporting the disastrous effects of stress on cognition, there are also research reports that show that they do not have any influence on general cognition in older adults (e.g. Giannouli, V., & Tsolaki, M. (2023). Stressful life events, general cognitive performance, and financial capacity in healthy older adults and Alzheimer’s disease patients. neuropsychiatrie, 1-4.

Zhu X, Yan W, Lin X, Que J, Huang Y, Zheng H, Liu L, Deng J, Lu L, Chang S. The effect of perceived stress on cognition is mediated by personality and the underlying neural mechanism. Transl Psychiatry. 2022 May 12;12(1):199. doi: 10.1038/s41398-022-01929-7.

Shields, G. S. (2020). Stress and cognition: A user’s guide to designing and interpreting studies. Psychoneuroendocrinology112, 104475.

Shields, G. S., Sazma, M. A., McCullough, A. M., & Yonelinas, A. P. (2017). The effects of acute stress on episodic memory: A meta-analysis and integrative review. Psychological bulletin143(6), 636.

Schwabe, L. (2017). Memory under stress: from single systems to network changes. European Journal of Neuroscience45(4), 478-489.). Authors should also discuss this body of evidence at the beginning of their article and then proceed with a more specific review of the literature.

Author Response

Reviewer 2’s original comments:

Comment 1: The procedure is not clear to the reader as well as the included material, so please describe in more detail.

Response: We added more details and a new figure (Figure 2) of the procedure of the experiment on pp.7-8.

Comment 2: In addition to that, the introduction and discussion sections need more references to support their points. Although there is a plethora of research supporting the disastrous effects of stress on cognition, there are also research reports that show that they do not have any influence on general cognition in older adults … Authors should also discuss this body of evidence at the beginning of their article and then proceed with a more specific review of the literature.

Response: We appreciate the Review for suggesting the additional literature. We have included the suggested articles in the revised manuscript. Specifically, the null findings of the effects of stress on cognitive were reviewed at the end of the first paragraph on p.1. Moreover, those conflicting findings were discussed with the present results in the Discussion section (pp.12-15).

Reviewer 3 Report

The manuscript entitled “Comparison between the effects of acute physical and psychosocial stress on feedback-based learning” describes a study that makes a noteworthy contribution to the literature on the effects of stress on cognitive functioning. The study is methodologically sound and its results are likely to be of interest to a broad segment of scholars and clinicians.

Beginning on Line 106, the authors state each of several hypotheses. If a brief rationale is attached to each hypothesis, the reader may be more prepared to agree with the methodological choices described in the section that follows. Simply put, a rationale will remind the reader that each hypothesis is accompanied by a sensible justification.

The authors state that “[a] modified probabilistic selection task was used to examine feedback-based learning”. Yet, they fail to explain why this task was chosen over other learning tasks. Similarly, the authors state that “[a] mental arithmetic task was used as an acute active-coping psychosocial stressor”. What is the rationale for choosing either task (including benefits) over others within the same task category?

The discussion section is limited, leaving the readers to question the relationship between the findings of the current study and earlier ones.  Specifically, to what extent the current findings may apply to other cognitive measures? To what extent do the current findings of no difference in the effects of physical and psychosocial stressors on feedback-based learning apply to other types of stressors and forms of learning?  A broader range of clinical applications may also be considered.

Author Response

Reviewer 3’s original comments:

Comment 1: Beginning on Line 106, the authors state each of several hypotheses. If a brief rationale is attached to each hypothesis, the reader may be more prepared to agree with the methodological choices described in the section that follows. Simply put, a rationale will remind the reader that each hypothesis is accompanied by a sensible justification.

Response: We added the rationale for developing hypotheses in the first paragraph on p.4.

Comment 2: The authors state that “[a] modified probabilistic selection task was used to examine feedback-based learning”. Yet, they fail to explain why this task was chosen over other learning tasks. Similarly, the authors state that “[a] mental arithmetic task was used as an acute active-coping psychosocial stressor”. What is the rationale for choosing either task (including benefits) over others within the same task category?

Response: The probability selection task was selected based on Frank et al. (2004), and we now elaborated on the reason why the task was used in the present study in the first paragraph on p.5. Moreover, we justified (or “added a justification for") the use the mental arithmetic task (MAT) and the comparison of the MAT with other similar tasks in the second paragraph on p.3 and the first paragraph on p.7. Please also see the response to Review#1’s Comment 1.

References:

Frank, M. J., Seeberger, L. C., & O’Reilly, R. C. (2004). By carrot or by stick: Cognitive reinforcement learning in parkinsonism. Science, 306, 1940–1943. https://doi.org/10.1126/science.1102941

Comment 3: The discussion section is limited, leaving the readers to question the relationship between the findings of the current study and earlier ones.  Specifically, to what extent the current findings may apply to other cognitive measures? To what extent do the current findings of no difference in the effects of physical and psychosocial stressors on feedback-based learning apply to other types of stressors and forms of learning?  A broader range of clinical applications may also be considered.

Response: We thank the Reviewer for the comment. We have expanded the Discussion section, supplied with additional literature, and elaborated on the implications of our findings on pp.11-15.